# Case-Control Study on the Routes of Transmission of SARS-CoV-2 after the Third Pandemic Wave in Tuscany, Central Italy

**DOI:** 10.3390/ijerph20031912

**Published:** 2023-01-20

**Authors:** Miriam Levi, Giulia Cereda, Francesco Cipriani, Fabio Voller, Michela Baccini

**Affiliations:** 1Epidemiology Unit, Department of Prevention, Central Tuscany Health Authority, 50135 Florence, Italy; 2Department of Statistics, Computer Science, Applications (DISIA), University of Florence, 50134 Florence, Italy; 3Tuscany Regional Centre for Work-Related Injuries and Diseases (CeRIMP), 50135 Florence, Italy; 4Epidemiology Unit, Regional Health Agency of Tuscany, 50141 Florence, Italy

**Keywords:** COVID-19, SARS-CoV-2, routes of transmission, case-control study, ridge regression, multiple imputation

## Abstract

The emergence of hyper-transmissible SARS-CoV-2 variants that rapidly became prevalent throughout the world in 2022 made it clear that extensive vaccination campaigns cannot represent the sole measure to stop COVID-19. However, the effectiveness of control and mitigation strategies, such as the closure of non-essential businesses and services, is debated. To assess the individual behaviours mostly associated with SARS-CoV-2 infection, a questionnaire-based case-control study was carried out in Tuscany, Central Italy, from May to October 2021. At the testing sites, individuals were invited to answer an online questionnaire after being notified regarding the test result. The questionnaire collected information about test result, general characteristics of the respondents, and behaviours and places attended in the week prior to the test/symptoms onset. We analysed 440 questionnaires. Behavioural differences between positive and negative subjects were assessed through logistic regression models, adjusting for a fixed set of confounders. A ridge regression model was also specified. Attending nightclubs, open-air bars or restaurants and crowded clubs, outdoor sporting events, crowded public transportation, and working in healthcare were associated with an increased infection risk. A negative association with infection, besides face mask use, was observed for attending open-air shows and sporting events in indoor spaces, visiting and hosting friends, attending courses in indoor spaces, performing sport activities (both indoor and outdoor), attending private parties, religious ceremonies, libraries, and indoor restaurants. These results might suggest that during the study period people maintained a particularly responsible and prudent approach when engaging in everyday activities to avoid spreading the virus.

## 1. Introduction

The COVID-19 pandemic is now in its third year; however, an in-depth knowledge of the most frequent modes of transmission of the SARS-CoV-2 virus needs to be acquired. The vaccination campaign that started at the beginning of 2021 was found to be effective in limiting COVID-19-related complications and deaths [1]. Nonetheless, the emergence of hyper-transmissible SARS-CoV-2 variants that rapidly became prevalent throughout the world in 2022, such as the Omicron variant and its sub-variants, made it clear that extensive vaccination campaigns cannot represent the sole measure to stop COVID-19. In fact, it is likely that for the foreseeable future, as the threat of new variants with increased transmissibility and/or virulence remains a concern, containment measures such as physical distancing, face mask use in closed or crowded spaces, frequent room ventilation, and hand hygiene will be required as tools able to reduce the contagion [2,3].

The effectiveness of control and mitigation strategies such as the closure of non-essential businesses and services, is debated [4,5]. Additionally, while some studies showed that, at least in the pre-Omicron phase, school closures were associated with a decrease in COVID-19 incidence and mortality [6], other studies concluded the contrary [7]. At the international level, only a few observational studies have been conducted to support decisions about COVID-19 containment measures [8,9,10]. These studies evaluated actions, places, and habits that may favour the virus spread, and found some evidence that participating in social events and going to restaurants or bars were associated with a higher incidence of infection.

In order to contribute to this literature, we conducted a questionnaire-based case-control study in the Tuscany region, central Italy, from May to October 2021, collecting information about individual behaviours and risks of infection. During most of the study period, Tuscany was classified as a medium/low risk area, according to the rules introduced by the ministerial decree issued on 3 November 2020 (https://www.trovanorme.salute.gov.it/norme/dettaglioAtto?id=76993), with the consequent relaxation of many restriction measures. Thus, we observed the impact of behaviours and public places attendance on COVID-19 infections in the first months after the easing of lockdown and the strictest epidemic prevention measures.

## 2. Materials and Methods

### 2.1. Data and Design

We conducted a study based on an online questionnaire to investigate potential differences in the places attended in the seven days prior to the COVID-19 diagnostic test by cases (i.e., those who tested positive for SARS-CoV-2) and controls (i.e., those who tested negative).

Data collection started on 2 June and finished on 10 October 2021, and retrospective answers were allowed. We focused on individuals who presented at the regional COVID-19 test sites from 1 May, assuming that participants accurately remembered events that had occurred up to one month before.

Participation in the survey was on a voluntary basis. Individuals were requested to avoid answers regarding COVID-19 tests performed in order to end the isolation consequent to a previous positive test result. At the time of the nasopharyngeal swab, each potential participant was handed out by the nursing staff a flyer containing the survey invitation, the description of the objectives of the study, as well as the link and the QR code to access the questionnaire itself in a dedicated web page in the Regional Health Agency website (https://www.ars.toscana.it/sars_cov2/questionario.php, accessed on 16 January 2023). The initiative was advertised through posters affixed at the COVID-19 test centres and via traditional and social media as well. Participants were asked to complete the questionnaire after receiving the notification of the test result from the healthcare authority, regardless of the result itself. For children under the age of 14, a parent or a legal guardian could fill in the questionnaire on behalf of the child. Children aged 14 to 18 were asked to complete the questionnaire on their own.

Data were collected anonymously. Relevant information gathered included the test result, place and date of COVID-19 test (self-reported by the respondent), demographics, living and work environments, places attended up to seven days before the test, face mask use, hand washing, and vaccination status. Additionally, those with a positive test result were asked to indicate the setting where in their opinion the contagion occurred.

The English translation of the survey, which was administered in Italian, is available in the Appendix A. The questionnaire did not contain any question that could allow researchers to identify and trace the respondents in the administrative healthcare databases.

The study setup and the questionnaire in Italian were approved by the Ethics Commission of the University of Florence on 4 May 2021 (Protocol code 0134393).

### 2.2. Statistical Analyses

#### 2.2.1. Multiple Imputation of Missing Data

A multiple imputation was performed to fill-in missing values in the data set. We generated M = 10 imputed data sets under the Missing at Random assumption, by implementing multiple imputation via chained equations (MICE) [11]. For each variable with missing values, the MICE procedure defined a regression tree (for quantitative variables) or a classification tree (for categorical variables) as the conditional model given all the other variables in the data set [12]. The multiple imputation considered the complete set of information obtained from the questionnaires.

#### 2.2.2. Logistic Regression

In order to estimate the association between each potential risk factor and the test result, we first specified single logistic regression models for every risk factor (one-factor-at-a-time analysis), with the test result as the outcome and the risk factor as the exposure, adjusting for the following fixed set of confounders: sex (males or females); age, categorised in the following age groups: 0–20, 20–40, 40–60 and 60+ years; vaccination status (dichotomized as “yes” if at least one dose of a COVID-19 vaccine had been received vs. “no”); calendar month when the COVID-19 test was performed (from May to October 2021); the Local Health Care Unit of residence (in Tuscany there are three units: Azienda USL Toscana centro, Azienda USL Toscana nord-ovest and Azienda USL Toscana sud-est); the size of the municipality of residence (<5000; between 5000 and 50,000; and >50,000 inhabitants). We used the *p*-values as scores to order the factors from the most to the least relevant, without any strict interpretation in terms of test hypothesis.

#### 2.2.3. Ridge Logistic Regression

We estimated the association between the outcome and each risk factor net of all others, by specifying a ridge logistic regression model [13]. In addition to the confounders, the ridge regression model included all the risk factors simultaneously, with a penalty on their coefficients to reduce problems deriving from collinearity between predictors. With the ridge regression, coefficient estimates are shrunken toward the null of an amount which is tuned by the penalty parameter. In our analysis, the value of the penalty parameter was selected by repeating 100 times a 10-fold cross-validation, then choosing the parameter associated with the lowest average of the 100 mean cross-validated errors [14].

#### 2.2.4. Combination of Results from Multiple Imputed Data Set

All the analyses were conducted separately for each of the M = 10 imputed data sets. Then, we used the Rubin’s rule to combine the 10 results thus obtained [15]. Specifically, in the case of logistic regression, we reported the estimate of the risk factor coefficient and its 90% confidence interval. For the ridge regression, we reported the estimates of the risk factors coefficients arising from the Rubin’s rule, with an evaluation of the between-imputation variability. We did not consider the within-imputation variability to avoid an over-interpretation of the estimated ridge regression standard errors that, as well-known, do not account for the fact that introducing a penalty brings a certain degree of bias in the estimates. For this reason, the results of the ridge regression should be interpreted in a qualitative rather than in a quantitative way, to obtain information about which risk factors are more predictive of the outcome, other things being equal.

#### 2.2.5. Sensitivity Analyses

The described procedures were first performed on the entire data set, and subsequently repeated on the subjects over 20 years of age to allow a more direct comparison with the literature [8,10]. We also produced results on the subjects who declared to not have had known contact with infected individuals in the week preceding the test.

We performed also analyses (i) by excluding respondents who had missing test results, and (ii) by excluding those who provided an inconsistent answer about the reason for testing (control after positivity, even though this was clearly indicated as an exclusion criterion in the introduction to the questionnaires).

#### 2.2.6. Evaluation of the Predictive Performance of the Ridge Regression Model

The predictive performance of the ridge logistic regression model was assessed by randomly splitting the data set into two parts: a training set, made of 95% of the participants (corresponding to 418 respondents), and a validation set, made of the remaining 5%. Prediction on the validation set was compared with the observed outcome and the AUC was calculated and averaged over the 10 imputed data sets. The procedure was repeated 10 times, and the overall average AUC was calculated. The cross-validation scheme is illustrated in Appendix A.

All analyses were performed using the R software [16]. The multiple imputation was performed by using the *mice* function of the *mice* package [17]. Ridge regressions were estimated by using the functions in the *glmnet* package [18].

## 3. Results

We analysed a total of 440 questionnaires. The database on which we performed the multiple imputation contained 5% of missing values. The summaries of the variables used for the statistical analyses (evaluated over imputations), with the corresponding number of missing values are reported in the Appendix A. The maximum number of missing values was observed for the variables swimming pool and seaside resort. In Appendix A, the number of positive cases per each risk factor is reported.

The result of the test was negative for 73.6% (n = 324) of study participants, positive for 15.9% (n = 70), and missing for 10.5% (n = 46). The majority of respondents were female (62%), and 80% of all participants were over 20 years of age. The percentage of participants who had received at least one dose of vaccine before the test was 61%.

From the one-factor-at-a-time analysis performed on the entire data set (Table 1), going to night clubs emerged as a factor strongly associated with a higher risk of infection, with a 90% confidence interval for the OR that did not include the null hypothesis of OR = 1 (OR 4.65, 90% CI: 1.62, 13.33).

Working with patients in health care settings (OR 1.80, 90% CI: 0.75–4.30), using crowded public transports (OR 1.72, 90% CI: 0.73, 4.06), and attending crowded clubs or outdoor bars at least once (OR 1.44, 90% CI: 0.81, 2.55) appeared to be associated with a higher risk of infection, even if in these cases the confidence intervals included the null value. Several behaviours appeared to be associated with lower risk of infections; those for which the 90% confidence interval of the OR did not include the null hypothesis were the following: visiting and hosting friends (OR 0.35, 90% CI: 0.20, 0.55; OR 0.35, 90% CI: 0.21, 0.58), attending open air theatres (0.17, 90% CI: 0.05, 0.61), performing sport activities (indoor: OR 0.24, 90% CI: 0.07, 0.91; outdoor: OR 0.40, 90% CI: 0.2, 0.81), participating in ceremonies (OR 0.25, 90% CI: 0.09, 0.73), going to small stores for shopping (OR 0.59, 90% CI: 0.37, 0.93), visiting hospitals (OR 0.51, 90% CI: 0.28, 0.93), and going to hair or beauty salons (OR 0.51, 90% CI: 0.27, 0.97).

The behaviours associated with a higher risk of infection among people over 20 years of age were the same observed on the entire data set: attending nightclubs (OR 6.23, CI 1.64, 23.72), working with patients in healthcare settings (OR 2.63, CI 1.03, 6.68), using crowded public transportation (OR 1.78, CI 0.64, 4.93), going to outdoor bars (OR 1.51, CI 0.87, 2.63), attending sport events (OR 1.40, CI 0.46, 4.25), and going to crowded clubs (OR 1.25, CI 0.64, 2.43). Focusing for brevity only on the ORs with a confidence interval that did not include the one, negative associations were observed for hosting and visiting friends (OR 0.26, CI 0.15, 0.46; OR = 0.35, 90% CI: 0.20, 0.62), attending outdoor performances (OR 0.19, CI 0.05, 0.74), participating in ceremonies (OR 0.31, CI 0.1, 0.94), and going to small stores for shopping (OR 0.57, 90% CI: 0.34, 0.96).

When we focused on the respondents over 20 years of age or on the subjects declaring to have had no known contacts with infected people, the results of the one-factor-at-a-time analysis were quite similar (Table 1).

The results of the ridge regression on the entire data set are reported in Figure 1.

The factors most associated with an increase in the infection risk were attending nightclubs, working with patients in healthcare settings, spending time in open-air bars, and to a lesser extent, going to open-air restaurants and crowded clubs, attending outdoor sporting events, and using crowded public transportation. Many factors resulted in being negatively associated with infection. Among them is, besides face mask use, attending open-air shows and sporting events in indoor spaces (even though with high inter-imputation variability), visiting and hosting friends, attending courses in indoor spaces, performing sport activities both indoor and outdoor, attending private parties, religious ceremonies, libraries, and indoor restaurants.

The average AUC from the ridge regression was 0.79 (minimum and maximum AUC over the 10 imputations: 0.75, 0.83).

The same analysis restricted to people over 20 years of age (Figure 2) led to the identification of the same factors.

However, it is worth noting that in this case the most relevant negative associations were estimated for visiting and hosting friends. When considering only participants who had no known contact with a case, the factors that showed the highest negative association with infection were visiting and hosting friends, followed by engaging in indoor and outdoor sports activities. Interestingly, among the factors that showed a positive association with the occurrence of infection, having travelled on crowded public transportation was the one for which we estimated the highest OR (Figure 3).

The two sensitivity analyses conducted after excluding missing test results and respondents with inconsistent answers about the reason for testing, led to qualitatively similar results (Appendix A).

Table 2 describes the answers on the individual perception about the routes of contagion among the respondents with a positive test result. Most of the infected respondents declared to suspect contagion from non-cohabiting family, friends, acquaintances, from travelling, from attending bars/pub/restaurants, from household contacts, or work.

## 4. Discussion

The present paper shows the results of a case-control study conducted in Tuscany from May to October 2021, a few months after the start of the COVID-19 vaccination campaign. During the study period there was a lifting of the restrictive prevention measures adopted in the previous months to contain the COVID-19 epidemic. This allowed us to assess the association between several activities and places with the SARS-CoV-2 infection. Two risk scenarios alternated during the study period: a medium/low-risk scenario up until 20 June, when the region was classified as a “yellow zone”, and a low-risk scenario of “white zone” from 21 June onwards. Different restrictions were associated with the two risk scenarios [19]. In the “yellow zones” the following activities were allowed: in bars and restaurants consumer services with consumption at the table were permitted exclusively outdoors; shows reopened to the public in theatrical halls, concert halls, cinemas, live-clubs, and in other venues or spaces, even outdoors; however, they could be carried out exclusively if seats were pre-assigned and provided that the interpersonal distance of at least one metre was respected both by spectators who were not habitually cohabiting and by staff; movements between different Regions were allowed even towards regions with higher (“orange” or “red”) risk classification for those in possession of the COVID-19 vaccination certificate, otherwise movements were allowed only towards regions with the same risk level; up to four people were allowed to visit friends or relatives from 5 a.m. to 10 p.m. Mask-wearing was compulsory in crowded places even in outdoor spaces. In the lowest-risk “white zones”, most of these previous restrictions were dropped, e.g., the limit on the guests’ number in private homes or the indoor dining restrictions in restaurants; mask-wearing rules were relaxed but remained mandatory in enclosed spaces, including schools, and until the end of July 2021 working from home was preferred for anyone who could work remotely.

Both the one-factor-at-a-time analysis and the ridge regression provided qualitatively similar results. These results suggest that people who engaged in recreational activities involving contact with many people, attending nightclubs, discos, pubs as well as outdoor venues where preventive measures such as mask-wearing and interpersonal distance could be relaxed, in the week before the test or the symptoms onset had a higher risk of testing positive. This finding is in line with the results of studies that investigated the transmission routes of COVID-19 cluster infections during the previous infection waves. For example, the role of recreational activity in infection spreading has been highlighted in Wong et al. [20], Lim et al. [21], and Liu et al. [22].

Regarding the use of public transportation, which has been reported by several studies as a circumstance causing the transmission of COVID-19 [22], our results on the entire data set and on people over 20 years of age provided some weak evidence of a higher risk of infection among people who declared having travelled on crowded trains or buses. This association appeared clearer from the analysis of the subset of subjects that denied contact with known cases in the week preceding the test or symptoms onset.

Being a healthcare worker in direct patient contact also emerged from our analysis as a possible risk factor for infection. However, considering that, in contrast, hospital attendance did not appear to be associated with higher test positivity, this result may simply reflect the care taken by health care workers in checking their own infection status by undergoing routine screenings, which are likely able to reveal infections that would otherwise remain undetected.

Notably, while only few factors resulted to be positively associated with the result of the SARS-CoV-2 test, many factors among those explored appeared to be associated with a reduced risk of infection: gathering with friends and families in private spaces, attending indoor bars and restaurants, as well as practicing sports and cultural activities, attending libraries, and participating in ceremonies. These results are somewhat in contrast with that perceived by individuals from the sample with a positive test result (Table 2) and from findings from previous studies, in which eating and drinking on-site in an indoor space were identified as the most important risk factors associated with SARS-CoV-2 infection [8,10,21]. The numerous negative associations estimated from the models suggest that the population adopted careful and virtuous behaviours when finally allowed to resume normal life, suspended for over a year on account of the COVID-19 pandemic. In order to safely engage in social activities and meeting friends or relatives, individuals adopted very cautious behaviours that allowed them to stay safe and protect others as well. They might have practiced self-isolation when feeling sick and met with others only if they felt well and in respect of the recommendations in place, such as face mask use, physical distancing, hand hygiene, and/or only after being fully vaccinated against COVID-19. Likely, pandemic fatigue, the psychological exhaustion that makes it difficult to maintain high the motivation required to adhere to recommendations [23] at the population level, had not yet really ensued. It is worth noting that these virtuous habits could have been mostly implemented when people planned to meet acquaintances, friends and family, which might explain why meeting friends and attending ceremonies appeared to be protective.

The negative association between sports practice and infection can be explained by the fact that people regularly engaged in sports activities are more likely to take measures to improve their health, thus also measures to prevent SARS-CoV-2 infection.

Regarding schools, our study did not provide solid evidence, also because school activities were reduced or interrupted during part of the study period. However, the sign of the relationship is in line with Hobbs et al. [9], who found that attending school was not associated with infection onset, and highlights the benefit of wearing face masks in schools.

The fact that the results did not change when removing subjects with missing test results and subjects providing an inconsistent response to the answer about the reason for testing is indicative of overall robustness of our conclusions.

In interpreting the results, we cannot rule out a certain degree of bias in the results related to the fact that participation in the survey was voluntary, and people could decide to respond after obtaining the test result. Perhaps, in the case of a positive test, some people who had engaged in behaviours that violated COVID-19 restrictions or could be considered risky (e.g., hosting or visiting friends without complying with constraints on the number of people) may have decided not to participate in the survey. This self-selection may have led to the excess of seemingly protective factors that emerged from the models. The possible presence of selection bias may have an effect also on the generalizability of the results to the entire population [24]. We might in fact speculate that those who took part in the survey were precisely those more likely to engage in healthy behaviours and that the reduced risk associated with certain activities and circumstances may be due to something similar to the so-called healthy user effect [25]. This would explain the low number of factors positively associated with test positivity. Additionally, even though the informational material urged people who were not very familiar with the use of computers or cell phones to obtain help in filling out the online questionnaire, some subgroups of the population may have declined to participate for reasons related to the type of questionnaire administration.

Part of the problems related to the selection of the participants can be avoided by adopting a different study design. For example, participants can be required to answer the items about their behaviours at the swab collection centre before or immediately after the sampling, then obtain the test result through administrative registers and link it to the questionnaire. This however would require the collection and handling of identification data, as well as considerable monetary and human resources to manage face-to-face interviews or self-administration of paper questionnaires distributed by the staff at the testing sites. Considering the contingent situation, namely the scarce resources available during the study period and the difficulty in rapidly obtaining the ethical approval to access individual test results, the collection of anonymous data through an online self-administered questionnaire was deemed the preferred option.

As a final remark, we want to stress that in comparing our results with the ones from studies conducted elsewhere and/or during different epidemic waves, one should consider that they refer to the specific epidemiological situation observed during the study period, when Delta was the most reported SARS-CoV-2 variant. With the emergence of more transmissible variants, such as the Omicron variant (Omicron found to be between 2 and 4 times more transmissible than Delta [26,27]) and sub-variants, the role of the different factors should be re-considered. Finally, the results also depend on the policies adopted during the study period. Thus, behaviours that did not emerge as positively associated with infection under the restrictions in place during the study period might emerge in their absence. Conversely, negative associations might disappear in the presence of relaxation or decreased adherence to restriction measures.

## 5. Conclusions

Although the results of our research must be interpreted with caution because of possible participant selection phenomena, in line with results from previous studies, we found that engaging in recreational activities that entail coming into contact with many people such as attending nightclubs, discos, pubs or outdoor venues, where the respect of mask-wearing and interpersonal distance could not be guaranteed, posed a higher risk of infection. Working in healthcare also was associated with a higher risk. In addition, we found evidence, albeit weak, of a higher risk of infection among people who travelled on crowded trains or buses. Nonetheless, in contrast with previous research, gathering with friends and families in private spaces, attending indoor bars and restaurants as well as practicing sports and cultural activities, attending libraries, and participating in ceremonies appeared to be associated with a reduced risk of infection. These results testify that, even in the absence of the strictest coronavirus containment measures, individuals can avoid contagion by adopting careful and virtuous behaviours when engaging in daily social activities. Our results also suggest an association between regularly engaging in sports activities and healthier habits and behaviours.

## Figures and Tables

**Figure 1 ijerph-20-01912-f001:**
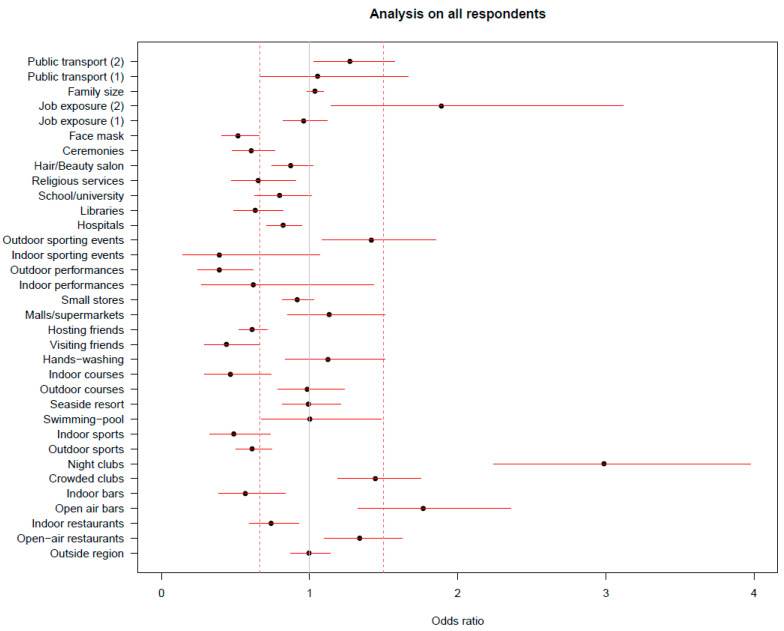
Results from the ridge regressions on the entire data set. Estimate of the odds ratio of positivity for each risk factor, adjusted for all the other ones and for the confounders; the reported segments represent the 90% confidence intervals based only on the between-imputation variability.

**Figure 2 ijerph-20-01912-f002:**
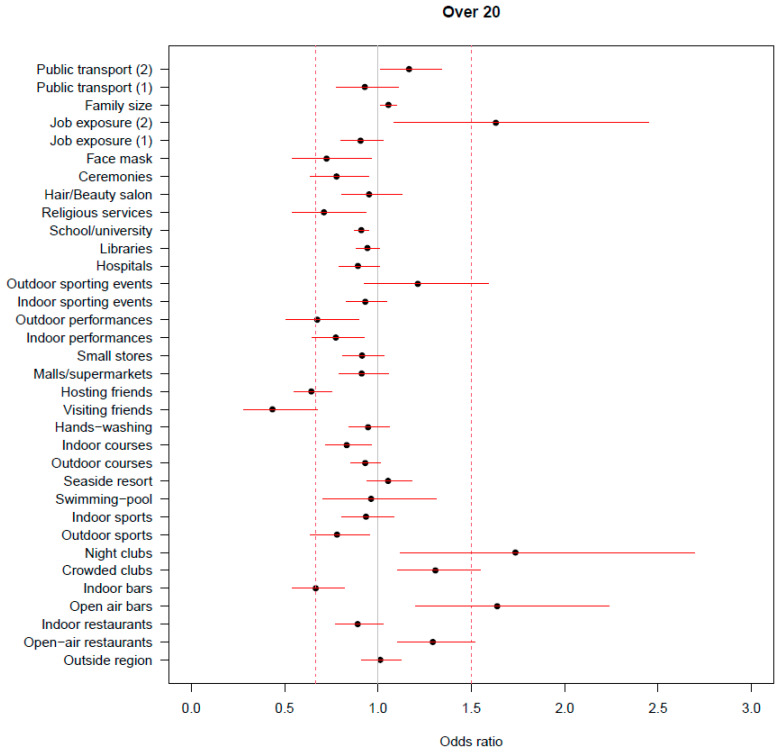
Results from the ridge regressions on participants over 20 years of age. Estimate of the odds ratio of positivity for each risk factor, adjusted for all the other ones and for the confounders; the reported segments represent the 90% confidence intervals based only on the between-imputation variability.

**Figure 3 ijerph-20-01912-f003:**
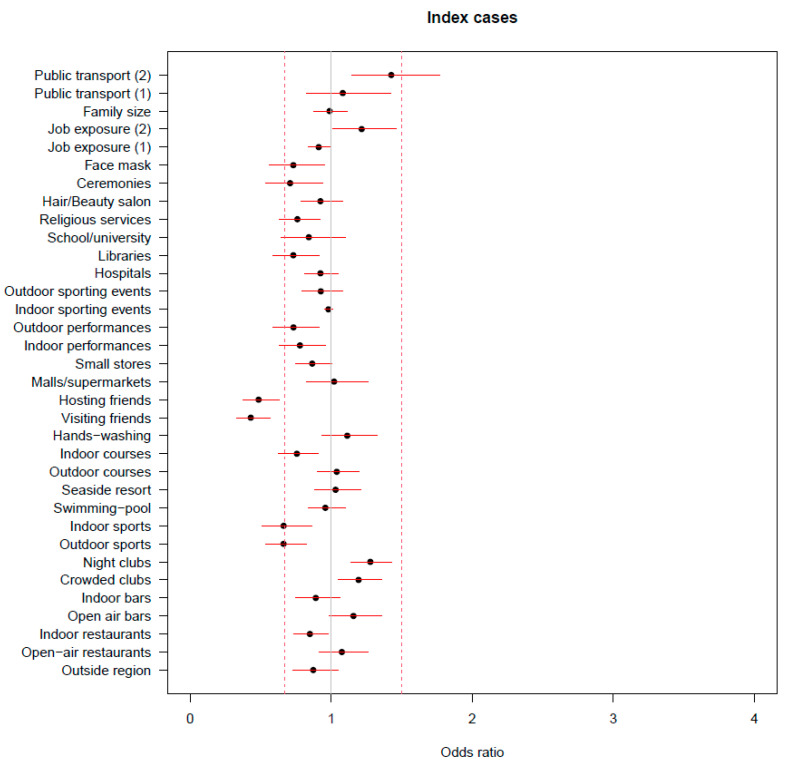
Results from the ridge regressions on the subset of participants with no known contact with infected individuals. Estimate of the odds ratio of positivity for each risk factor, adjusted for all the other ones and for the confounders; the reported segments represent the 90% confidence intervals based only on the between-imputation variability.

**Table 1 ijerph-20-01912-t001:** Results from the logistic regressions on the entire data set, on the subset of subjects with no known contact with infected individuals, and on those over 20 years of age. Estimate of the odds ratio (OR) for each risk factor *, adjusted for the confounders, 90% confidence interval (LB: lower bound; UB: upper bound), and *p*-value.

	Analysis on All Respondents				Index Cases (without Known Contacts with Positives)				Over 20			
	Adjusted OR	LB 90%	UB 90%	*p*-Value	Adjusted OR	LB 90%	UB 90%	*p*-Value	Adjusted OR	LB 90%	UB 90%	*p*-Value
Public transport (2)	1.72	0.73	4.06	0.301	2.14	0.74	6.16	0.237	1.78	0.64	4.93	0.356
Public transport (1)	1.03	0.41	2.60	0.961	1.03	0.32	3.30	0.966	0.95	0.38	2.37	0.920
Family size	1.05	0.88	1.25	0.650	1.03	0.82	1.29	0.841	1.06	0.89	1.27	0.588
Job exposure (2)	1.80	0.75	4.30	0.271	1.66	0.50	5.54	0.489	2.63	1.03	6.68	0.089
Job exposure (1)	0.88	0.49	1.59	0.730	0.95	0.45	1.99	0.907	0.96	0.51	1.80	0.914
Face mask	0.63	0.34	1.17	0.222	0.88	0.36	2.13	0.808	0.63	0.28	1.42	0.353
Ceremonies	0.25	0.09	0.73	0.034	0.14	0.02	0.79	0.062	0.31	0.10	0.95	0.085
Hair/Beauty salon	0.51	0.27	0.97	0.087	0.57	0.26	1.22	0.225	0.63	0.32	1.24	0.260
Religious services	0.48	0.22	1.07	0.132	0.38	0.12	1.17	0.158	0.47	0.20	1.15	0.166
School/University	0.48	0.20	1.12	0.156	0.50	0.15	1.62	0.333	0.61	0.19	1.91	0.477
Libraries	0.35	0.05	2.30	0.363	0.00	0.00	Inf	0.989	0.51	0.07	3.66	0.574
Hospitals	0.51	0.28	0.93	0.068	0.54	0.25	1.14	0.176	0.55	0.29	1.05	0.128
Sporting events	1.16	0.41	3.32	0.816	0.53	0.08	3.46	0.579	1.40	0.46	4.25	0.618
Outdoor performances	0.17	0.05	0.61	0.023	0.24	0.06	0.99	0.098	0.19	0.05	0.74	0.045
Indoor performances	0.33	0.07	1.52	0.234	0.23	0.03	1.52	0.201	0.29	0.06	1.38	0.194
Small stores	0.59	0.37	0.93	0.059	0.52	0.28	0.98	0.091	0.57	0.34	0.96	0.078
Malls/supermarkets	0.69	0.40	1.17	0.249	0.58	0.27	1.25	0.247	0.60	0.33	1.10	0.168
Hosting friends	0.35	0.22	0.58	0.001	0.19	0.09	0.39	<0.001	0.35	0.20	0.62	0.002
Visiting friends	0.33	0.20	0.55	0.000	0.17	0.09	0.35	<0.001	0.26	0.15	0.46	<0.001
Hands-washing	0.91	0.46	1.81	0.825	1.20	0.46	3.13	0.753	0.81	0.36	1.82	0.667
Courses	0.57	0.26	1.28	0.258	0.46	0.15	1.38	0.245	0.55	0.22	1.33	0.265
Seaside resort	0.97	0.54	1.75	0.928	0.79	0.35	1.76	0.625	0.90	0.47	1.72	0.796
Swimming-pool	0.92	0.38	2.25	0.878	0.59	0.18	1.89	0.456	0.70	0.22	2.21	0.608
Indoor sports	0.24	0.07	0.91	0.078	0.14	0.02	0.93	0.089	0.62	0.15	2.57	0.582
Outdoor sports	0.40	0.20	0.81	0.034	0.29	0.11	0.75	0.034	0.42	0.17	1.05	0.121
Night clubs	4.65	1.62	13.33	0.017	3.68	0.82	16.49	0.154	6.23	1.64	23.72	0.025
Crowded clubs	1.44	0.81	2.55	0.301	1.00	0.44	2.28	0.993	1.25	0.64	2.43	0.585
Indoor bars	0.45	0.20	1.01	0.104	0.56	0.22	1.45	0.316	0.44	0.19	1.01	0.104
Open air bars	1.38	0.84	2.26	0.284	0.82	0.41	1.62	0.628	1.52	0.87	2.63	0.215
Indoor restaurants	0.59	0.31	1.13	0.181	0.57	0.26	1.24	0.235	0.69	0.34	1.37	0.373
Open-air restaurants	1.03	0.64	1.68	0.911	0.74	0.39	1.40	0.434	1.13	0.65	1.97	0.720
Outside region	0.94	0.52	1.69	0.854	0.62	0.28	1.39	0.335	0.97	0.51	1.85	0.930

* The result for attending libraries is not reported due to perfect collinearity of the risk factor indicator with the confounders.

**Table 2 ijerph-20-01912-t002:** Summary description of the individual perception about the routes of contagion among the respondents with a positive test result.

Where Do You Think You Contracted the Virus? *	Number of Responses (% of all Who Tested Positive)
Household contacts	10
At work	10
At school	3
At the hospital/at the doctor’s cabinet	3
From non-cohabiting family members, friends, acquaintances	15
While carrying out sporting activities	2
While carrying out cultural activities	1
At the bar/restaurant/pub	10
In a shop/at a shopping mall	6
At the cinema/theatre	0
Ceremonies or religious functions	1
Travelling	11
I do not know	16
Other	7
NA	1

* More than one answer allowed.

## Data Availability

Raw data are freely available upon request to the corresponding authors.

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
