# Peer review of "Case-Control Study on the Routes of Transmission of SARS-CoV-2 after the Third Pandemic Wave in Tuscany, Central Italy"

_ijerph, 2023, doi:10.3390/ijerph20031912_

Round 1

Reviewer 1 Report

The study results are relevant and the scientific contribution is necessary for a better understanding of peoples' behavior impact on virus spread. I would recommend reviewing and presenting the methodology section in a more structured way, separating it into a few more paragraphs. Also, a visual presentation of the applied framework could give a better understanding. 

Reviewer 2 Report

The study by Levi et al assess the infection risk for SARS-CoV-19 for different exposure factors. The authors issued a questionnaire to individuals presenting for SARS-CoV-19 testing and then compared the answers and potential exposure sites to infection risk. The study data used 440 questionnaire outputs and concluded that ‘night clubs’ had the highest infection risk, while gatherings with friends at home had the lowest infection risk. The study also reported on different infection risks for attending outdoor and indoor sporting events, with the latter having a low risk but the former a high risk. This is an interesting study showing the safe practices of family/friends gathering and outdoor events compared to less safe gatherings in night clubs.

The methods seem sound, although, I do not have the required expertise to assess the algorithms used. I am not sure if odds ration can only be regarded important if significant.

However, the study is at times confusing and lacks overall consistency.

Major comments:

1.     Of the 440 questionnaires analysed, only 70 had a positive SARS-CoV-2 test results. Thus, it makes in challenging to estimate how many positives and negatives were in each category. For example, is the strong positive correlation for visiting night clubs based on links with index cases? As this category was not significant for those without index case (Table 1). Also, the authors report on >35 different risk factors, and seems too many with only 70 positive cases. How many positive cases are there per category?

2.     Also, some of the risk factors are overlapping (night clubs/crowded clubs or ceremonies/religious services) others are non-specific (outside region, family size). Table 2 then has different factors, such as school and cinema but is missing night club. More consistency throughout the manuscript would be useful in terms on risk factors analysed.

3.     The authors split the data set into 3 categories: all, without known index case, and >20 year old. The authors say that the use over 20 year old as a sub category to make their results more similar to the literature. However, the authors also say that there are no differences in risk outcome between the complete data set and the >20 year olds (which is not surprising as 80% of their data was in the latter category anyway). To simplify understanding I would suggest moving the results for ‘all participants’ into the supplementary and mention in the main text that there is no difference. Leave table 1 but remove the ‘entire data set results’ in the figure sections.’

Minor comments:

Abstract is missing an introductory sentence (Background).

Line 36. Replace ‘seems to have been effective’ with ‘was effective’

Line 39. Does ‘sub-variants’ refer to sub-variant for Omicron? The phrasing is a bit odd.

More correctly would be for example: ‘ such as the Omicron variant and its sub-variants’.

Lines 37 ff. The sentence is too long and should be split into 2 or 3 sentences.

Line 51.  Should be changed to ‘,and found some evidence’

Line 60 ff. the phrasing is a bit odd, please change.

Line 69. Add year

Line 83. Tutor seems an odd choice. I think ‘legal guardian’ is the appropriate term

Line 142. Should be changed to ‘over 20 years of age’ or ’20 years old’.

Line 162. 80% of all participants were over 20 or 80% of female participants were over 20? Please clarify and add ‘over 20 years of age or 20 years old. Do this whenever over 20 is mentioned.

Line 165. Should read: ‘Having gone to night clubs’ and ‘higher risk of infection’.

Line 173. I don’t understand this sentence, please rephrase.

Table 1. ‘Estimate’ should be changed to ‘odd ratio’.

Table 1 should be shown as a figure like figures 1-3. Show the 3 different groups on one figure in 3 colours, simplify comparison

Conclusion is too long.
